



# Research on Retrieval of Atmospheric Temperature and Humidity Profiles from combined Ground-based Microwave Radiometer and Cloud Radar Observations

Yunfei Che[1], Shuqing Ma[2], Fenghua Xing[3], Siteng Li[4], Yaru Dai[5]

[1] Key Laboratory for Cloud Physic, Chinese Academy of Meteorological Sciences, Beijing 100081, China
[2] Meteorological Observation Centre of China Meteorological Administration, Beijing 100081, China
[3] Hainan Institute of Meteorological Sciences, Haikou, Hainan 570203, China
[4] Beijing Municipal Meteorological Observation Center, Beijing 100089, China
[5] National Space Science Center, Chinese Academy of Sciences, Beijing 100029, China

Correspondence to: Shuqing Ma (cyfcuit@163.com)

**Abstract.** This paper focuses on the retrieval of temperature and relative humidity profiles through combining ground-based microwave radiometer observations with those of millimeter-wavelength cloud radar. The cloud-base height and cloud thickness from the cloud radar were added into the atmospheric profile retrieval process, and a back propagation neural network method was used as the retrieval tool.

Because substantial data are required to train a neural network, and microwave radiometer data are insufficient for this
purpose, eight years of radiosonde data from Beijing were used as a database. The model MonoRTM was used to calculate the brightness temperature of the same channel as the microwave radiometer. Part of the cloud-base height and cloud thickness in the training dataset was also estimated using the radiosonde data.

The accuracy of the results was analyzed by comparing with L-band sounding radar data, and quantified using the mean bias, root-mean-square error and correlation coefficient. The statistical results showed that inversion with cloud
information was the optimal method. Compared with the inversion profiles without cloud information, the RMSE values after adding the cloud information were to a varying degree reduced for the vast majority of height layers. These reductions were particularly clear in layers with cloud present. The maximum reduction of RMSE for temperature was 2.2 K, and for the humidity profile was 16%.

## 1 Introduction

Radiosonde observations are generally made no more than four times in one day, not only making it difficult to describe atmospheric processes in detail, but also inhibiting the possibility of achieving the accuracy required for researching and forecasting small and mesoscale severe weather systems.

Infrared spectrometers and radiometers carried by satellites can provide remotely sensed parameters of atmospheric profiles, and can also be used at a high temporal resolution; however, the vertical resolution and accuracy of remote
sensing techniques is limited when detecting near the ground (Wu et al., 2005).



Over the past 20 years, many researchers have shown that ground-based remote sensing is capable of measuring atmospheric profiles in the lower troposphere. Therein, the ground-based microwave radiometer (MWR) is an important instrument for retrieved humidity and temperature profiles. It can observe continuously, providing data on the complete process of an entire weather event as it evolves(Candlish et al., 2012). In addition, ground-based measurements are more suitable compared with those of satellites when detecting at heights of between 0 and 10 km.

Through a number of studies it has been determined that the performance of the MWR in retrieving atmospheric profiles is good under clear conditions. However, on the contrary, retrievals tend to be far worse under cloudy or rainy conditions, especially relative humidity profiles, with some extreme cases yielding acutely obvious errors(Chan et al., 2009). This problem is particularly pertinent given that cloudy or rainy conditions exist during most forms of disastrous weather, such as rainstorms, strong winds, hail and typhoons — and for obvious reasons these meteorological phenomena are the ones we are most interested in understanding and forecasting.

Thus, various researchers have attempted to construct a system that combines vertical observations in such a way to obtain more accurate atmospheric parameters. (Frate et al., 1998; Lijegren et al., 2001; Lohnert et al., 2001). For instance, Stankov et al.(1996) and Klaus et al., (2006) combined radar wind profiler and MWR measurements to estimate atmospheric humidity profiles. Brandau et al.(2010), Frisch et al.(1995) and Lohnert et al.(2001) evaluated the liquid water content of clouds through a combination of millimeter-wavelength radar and MWR data, demonstrating that significant improvements could be made in retrieving cloud liquid water profiles by adopting such an approach.

The MWR can measure normally in the precipitation-free cloudy conditions, and the drop deposition on the instrument may cause incorrect measurement of MWR in the precipitation conditions. However, the presence of cloud can lead to an obvious change in the brightness temperature (BT) measured by the MWR compared with clear-sky conditions. Meanwhile, relative humidity can increase rapidly in the cloud layer, and temperatures may also change. However, owing to the uncertainty caused by clouds, the performance of retrievals tend to be worse than clear-sky conditions.

The present study combines active and passive remote sensing, based on techniques presented previously in the literature, with the aim to improve retrievals by ground-based MWRs under cloudy conditions. Atmospheric profiles were detected through a combination of ground-based MWR and millimeter-wavelength cloud radar (MWCR) observations. The retrieval tool used was a back propagation neural network (BPNN), which is an important and mature technique (Churnside et al., 1994; Solheim et al., 1998). The cloud information taken from the cloud radar was added during the MWR atmospheric profile retrieval process, and the accuracy of the MWR retrieval was analyzed through comparison with L-band sounding radar data. The impact of including cloud information when retrieving atmospheric profiles could then be assessed.

## 2 Data sources and pre-treatment

### 2.1 Data sources

The data used in this study include radiosonde data, BT, the retrieval product of the MWR, and cloud-base height and





thickness from the MWCR. The radiosonde data, with a temporal resolution of 1s, were obtained from an L-band GTS1 digital radiosonde. The sounding balloon requires around 40 min to travel from 0 to 10 km above ground level. The MWR used in this study was the RPG-HATPRO model (Radiometer Physics GmbH). The RPG-HATPRO 14-channel ground-based MWR includes seven K-band frequency channels of between 22 and 30 GHz, and seven V-band channels of between 51 and 59 GHz. The absolute accuracy of the BT is 0.5 K.

The MWCR was contructed by the Meteorological Observation Centre of the China Meteorological Administration and Xi'an Huateng Microwave Co. Ltd. The cloud radar is a solid state Doppler radar oriented vertically and with a working frequency of 35 GHz, peak power of 4 W, and sounding range of 12 km. It has a spatial resolution of 30 m and an adjustable temporal resolution of between 1 and 60 s. Photographs of all of the instruments used are shown in Figure 1.

Ideally, the three instruments should make their observations at the same time and location, to eliminate system error as much as possible. However, the position of the detector differed slightly among the instruments because of limitations imposed by the experimental conditions. The distance between the MWR and the cloud radar was 61.75 m; the distance between the cloud radar and radiosonde was 162.37 m; and the distance between the MWR and radiosonde was 182.51m. The training dataset in this study was based on eight years (from 2006 to 2013) of annual radiosonde data. Soundings were made twice daily (11:15 and 23:15 UTC). Initially, the pre-treatment of the radiosonde data focused on the removal of rainy or uncertain weather conditions. There were 2715 training samples, of which 1626 were taken during clean-sky conditions and 1089 during cloudy conditions. The test dataset included 100 randomly selected groups of year 2006-2013 data, which were not used for training, and 382 groups of sounding data from 2014 to January 2015 (excluding rainy conditions). It should be noted that, because of the limited amount of cloud radar data, the cloud-base height and cloud thickness from the year 2013 to 2015 were provided by the MWCR and the cloud information of other samples was obtained by analyzing the relative humidity from the radiosonde.

## 2.2 Pre-treatment of sounding data

The training process required substantial cloud information, but not all samples could be matched to the information from the cloud radar, which had only been running since 2013; therefore, the radiosonde data were used to estimate cloud parameters in some cases. This section describes the method used to determine the weather conditions from the radiosonde sounding data. In theory, cloud should form when relative humidity reaches 100%, but because of various factors such as the presence of condensation nuclei, clouds can form when relative humidity reaches around 85%. This study used 84% as the threshold for relative humidity, following the work of Wang et al.(1995). The specific methods used to determine the weather conditions were as follows:

If the relative humidity was consistently less than 84% from the ground to any height, the weather was classified as clean-sky. If the relative humidity was consistently greater than 84% from the ground to a height of 600 m, the weather was classified as rain. If the relative humidity was less than 84% near the ground, but there was stratification with relative humidity greater than 84%, the weather was classified as cloudy. Under the cloudy conditions, the cloud-base height and cloud-top height were determined based on the work of Wang et al.(1995).



The above method was used to filter the radiosonde data by clean-sky and cloudy conditions and obtain the cloud-base height and thickness at the times of radiosonde observations under cloudy conditions.

## 2.3 Comparison of cloud determined by radiosonde and cloud radar

We used the cloud-base height and cloud thickness estimated by the radiosonde data to supply cloud information when cloud radar data were absent. However, to reduce the impact of errors from this estimated cloud information on the accuracy of the retrieval, we verified the feasibility of the method by comparing 60 groups of cloud information estimated by the radiosonde data with simultaneous MWCR measurements. The results of the comparison are shown in Figure 2.

In the sample statistics of the 60 groups of cloud information, there were five groups in which the clouds were shown to be present according to one instrument but not according to the other one. Beacause of the presence of condensation nuclei and ice crystal, the humidity might not be up to 84%, which could cause the missing of the cloud detection by radiosonde. And the cloud radar might sometimes miss the thin cloud within small cloud droplets.

Excluding these five groups of abnormal samples, the remaining 55 groups were used as statistical samples. The results of the comparison were as follows: For cloud-base height, the mean bias was 0.476 km and the correlation coefficient was 0.936. For cloud thickness, the mean bias was 0.450 km and the correlation coefficient was 0.723.

Though it is undeniable that the deviation existed between the cloud radar measurement and the cloud estimated by relative humidity from radiosonde this method can get the vertical distribution of cloud, the cloud information must be more accurate and richer than before. Through the quantified statistics, the deviation could be acceptable in this experiment. More correction of the cloud data should be inplemented in the follow-up work.

## 3 Retrieval methodology

### 3.1 Simulating BT based on MonoRTM

Because of the limitations on the number of measuring BT with the MWR, but knowing that substantial BT data should be added during the retrieval process, many researchers have used the radiative transfer equation of atmospheric microwave radiations to simulate the BT. Here, We calculated BT based on monochromatic radiative transfer model(MonoRTM) (Clough et al., 2005). The model was provided by Atmospheric and Environmental Research, Inc. and uses the same physics and continuum model as the line-by-line radiative transfer model (Clough et al., 1992). MonoRTM is suitable for the calculation of radiances associated with atmospheric absorption by molecules in all spectral regions, and cloud liquid water in the microwave region. The model uses the Humlicek Voigt Line Shape and the MT-CKD continuum (Mlawer et al., 2012) to handle molecular absorption not included in the "line center" of each spectral line.

The calculations of BT for the same channels of the ground-based MWR were obtained by applying MonoRTM to the radiosonde data for Beijing for the period 2006–2013. The sounding data were pre-processed according to the input requirements of MonoRTM. The data were divided into two weather cases: clean-sky and cloudy. For the clean-sky



conditions, the simulated BT could be obtained after input of the treated sounding profile into MonoRTM. For cloudy conditions, the model required the cloud liquid water content at the height of the cloud layer prior to the BT calculation. However, the required profiles of cloud liquid water content were not available from conventional upper-air ascent data. Therefore, in this study they were approximated using the relative humidity, following the method of Poore et al.(1995).

To verify the accuracy of MonoRTM, the BT results were compared with observations from the MWR. The BT in the 22.24 GHz and 58 GHz channels were consistent with observations, as shown in Figure 3.

The BT was simulated with 60 statistical groups compared to the measurements by the MWR. In the 60 samples, there were 18 samples in clear-sky cases, 42 samples in cloudy cases, it should be noted that the precipitation cloud and the cloud with very rich liquid water cases were not accounted. The accuracy was quantified by mean deviation (MD),

standard deviation (SD), and correlation coefficient ($\rho$). The statistical results are shown in Table 1.

The maximum MD value between the simulated BT and MWR measurements was 2.145 K, and the SD of the MD was 2.508 K. The correlation coefficient was greater than 0.9 for all channels. It is important to note that the radiosonde and MWR were not positioned in the same place, and the sounding balloon may have drifted with increased height. Nonetheless, despite the existence of these individual errors, the deviation of the statistical data was acceptable.

## 3.2 BPNN

The linear regression method and neural networks are probably the most popular in current research. Compared with the regression method, neural networks (Churnside et al., 1994; Frate et al., 1998) are considered to offer an improvement in terms of capturing the complicated nonlinear relationship between the BT and retrieved atmospheric profiles.

### 3.2.1 Principle

Crucially, there is no need to create a new and complicated algorithm because the BPNN method can in theory approximate any complex nonlinear relationship. Neural networks have been widely applied in the field of generating atmospheric parameters of the inversion profile. We used a three-layer BPNN, which can obtain any precision of a continuous function. A diagram of the BPNN is shown in Figure 4.

L is the number of elements in the input layer, M is the number of elements in the hidden layer and N is the number of

elements in the output layer. IW and LW are the weights of input layers to hidden layers and the hidden layers to the output layers. The bias value is represented by b.

A tansig transfer function is used from the input layer to the hidden layer. Its expression is:

$$\text{tansig(n)} = \frac{2}{1+\exp(-2*n)} - 1 \qquad\qquad (1)$$

Therefore, the relationship between the hidden layer and the input layer can be expressed as:

$$\text{HL} = \text{tansig}(\text{IW}\{1,1\} \cdot \text{IL} + \text{b}\{1\}) \qquad\qquad (2)$$

where HL is the vector of hidden layers, and IL is the vector of input layers.



The linear transfer function purelin is used in the hidden layers to output layers, giving the relationship as:

$$OL = purelin(LW\{1,1\} \cdot HL + b\{2\}) \tag{3}$$

where OL is the vector of output layers.

The weights and the bias are determined during the training process; they are obtained using the back-propagation algorithm that is described in detail in Rumelhart et al.(1986). This algorithm adjusts the weights and bias iteratively to reduce the difference between the actual training set output vectors and the estimated output vectors calculated by the network using the input vectors of the training set.

### 3.2.2 Methodology on comparison of the addition of cloud

To better analyze the impact of cloud parameters on the retrieval of atmospheric profiles, we fixed the other parameters and used the cloud base height and cloud thickness as the independent channels to input into the BPNN. The information was added to train the network and to build a new neural network model. The results of this new model were then compared with the retrieval without cloud.

In the inversion model without cloud information, the input vector comprised 17 elements. The first 14 elements were the BT in 14 radiometer channels. The 15th to 17th elements were surface temperature, relative humidity, and pressure. The output vectors were a 47-element vertical profile of temperature or relative humidity. The vertical resolution was 100 m between the heights of 0 and 1 km, and 125 m between 1 and 10 km. The number of elements in the hidden layer was selected according to Mirchandani et al.(1992).

$$M = \sqrt{0.42LN + 0.12N^2 + 2.54L + 0.77N + 0.35} + 0.51 \tag{4}$$

In the model of inversion with cloud information, the output vector was the same as the model without cloud information. However, the input vector included two more elements than the model without cloud. The first 17 elements were the same as the model without cloud, and the final elements were the cloud-base height and cloud thickness.

Using the two models described, all samples were trained by different methods to obtain two different parameters of the neural network, which had different input elements and the same output elements. The test sample should be input with the requirements of the input layer in different models.

### 3.3 RPG-HATPRO retrieval method

In order to estimate atmospheric profiles from radiometer data, the RPG-HATPRO radiometer has its own retrieval algorithm. The manufacturer's software provides three selectable retrieval types: linear regression, quadratic regression and neural networks. The retrieval type in this experiment was quadratic regression.

The quadratic regression retrieval calculation has the following structure:

$$Out_i = OS_i + \sum_{sensors} SL_{ij} * Sr_j + \sum_{freq} TL_{ij} * Tb_j + \sum_{sensors} SQ_{ij} * Sr_j^2 + \sum_{freq} TQ_{ij} * Tb_j^2 \tag{5}$$

where $Out_i$ is the i-th retreival output parameter, $OS_i$ is the retrieval offset parameter for $Out_i$, $Sr_j$ is the surface


sensor reading of the j-th checked sensor(the sequence is: temperature sensor, humidity sensor, pressure sensor and infrared radiometer). $SL_{ij}$ is the corresponding linear coefficient and $SQ_{ij}$ is the corresponding quadratic coefficient, $Tb_j$ is the brightness temperature at the j-th frequency with $TL_{ij}$ is the corresponding linear coefficient and $TQ_{ij}$ is the corresponding quadratic coefficient.

The quadratic regression retrieval process also used nearly ten years of sounding data as the database, but the sounding data used in the RPG retrieval were provided by the university of Wyoming (online at http://weather. uwyo.edu/upperair/sounding.html). The corresponding simulating radiometer BT data also need to be calculated with MonoRTM. Through the relationship of the calculating BTs, sensor data and the sounding data, the coefficients of the quadratic regression was obtained. And the coefficients are applied to the radiometer measurements in order to retrieve
the actual atmospheric variables.

## 4 Influence of cloud on retrieval

### 4.1 Theory

When considering a plane parallel atmosphere, scattering can be ignored and the zenith angle at the direction of radiative transfer is given as $\theta$. For ground-based remote sensing, the atmospheric downward radiance that is received can be
expressed as Tan et al.(2011):

$$T_{B\beta}^{\downarrow}(\theta,0) = T_{B\beta}(\infty)\tau(0,\infty) + \int_0^\infty k_\alpha(z)T(z)\tau(0,z)sec\theta dz \qquad (6)$$

In the above equation, $T_{B\beta}(\infty)\tau(0,\infty)$ refers to the cosmic radiation in the background after attenuation in the atmosphere, in which the quantity $T_{B\beta}(\infty)$ is the cosmic background temperature,usually taken to be 2.75 K; $\tau(0,z)$ is the transmittance from height z to the ground; $T(z)$ is the atmospheric temperature at the height z, and $k_\alpha$ is the
20 atmospheric absorption coefficient. In clean-sky conditions, the absorption coefficient mainly results from absorption by oxygen molecules and water vapor. It can be expressed by: $k_\alpha = k_{O_2} + k_{H_2O}$. However, in cloudy conditions, the calculation of the atmospheric absorption coefficient is different. The cloud layers, including cloud liquid water, impact upon the absorption coefficient substantially. The absorption coefficient can be expressed as: $k_\alpha = k_{O_2} + k_{H_2O} + k_{cloud}$. The existence of cloud layers can change the BT measured by the MWR. Research has shown that differences in cloud
height, cloud thickness and density of cloud liquid water have different effects on the measurement of BT. However, the provision of cloud physical parameters by the MWR is poor during the process of retrieving atmospheric profiles. To obtain cloud information, the MWR itself is configured with far infrared radiometer components, which can measure the infrared radiation brightness temperature of the sky to judge the existence of cloud and estimate the cloud base height. However, because of the impact of many factors such as the atmosphere, aerosols and cloud structure, the cloud-base
height measured by infrared sensors requires error testing and it is difficult to obtain accurate data in the long-term. Moreover, not only cloud-base height, but also cloud thickness and the distribution of cloud liquid water, and even particles in cloud may have a considerable impact on the retrieval. Because the information that the retrieval process





requires is lacking, the inversion of temperature and humidity profiles may produce large error in cloud estimations. Therefore, we combined of active and passive remote sensing and built a joint observation system that included both the MWCR and MWR. In addition, the more accurate cloud information from the cloud radar was added into the process of retrieving the atmospheric profiles using the BPNN as the key tool.The accuracy of the retrieval results was analyzed through comparison with L-band sounding radar data. Based on the above approach, the impact of the inclusion of cloud information on the retrieval of atmospheric profiles could be assessed.

## 4.2 Typical cases

In this section, the radiosonde data as standard are compared with the BPNN retrieval profiles with and without cloud and the RPG retrieval result. To compare the results of the three retrieval methods and the radiosonde more intuitively, we present four typical cases of temperature(Figure 5) and relative humidity(Figure 6) profiles at different times under cloudy conditions.

As can be seen from the four typical cases, the three retrieved temperature profiles(Figure 5) are relatively close to the sounding from the ground to 5 km. However, deviation of the RPG product increases considerably at heights above 5 km, and the difference between BPNN(No-cloud) values and those from the radiosonde increases above 8 km. Although BPNN(Cloud) also shows deviation from the radiosonde data above 9 km, the deviation is clearly lower than that for the other two methods.

For relative humidity(Figure 6), the retrievals of different methods are far more variable than those for temperature. The MWR can approximately follow the patterns of the radiosonde data with the change of height, but the deviation with increases clearly with the inclusion of cloud, and the trend with increasing height is not the same as in the sounding data. In addition, the bias of BPNN(No-cloud) is large above the cloud height, but this is substantially improved after adding the cloud information.

## 4.3 Statistical analysis

The accuracy of the retrieved atmospheric profiles was quantified statistically by comparison with the actual radiosonde measurements in terms of the mean bias (MB), root-mean-square error (RMSE), and correlation coefficient($\rho$), as follows:

$$\text{MB} = \frac{1}{n}\sum_{i=1}^{n}|R_i - Q_i| \tag{7}$$

$$\text{RMSE} = [\frac{1}{n}\sum_{i=1}^{n}(R_i - Q_i)^2]^{1/2} \tag{8}$$

where n represents the number of samples, i indicates the i-th sample, $R_i$ represents the retrieved temperature and relative humidity at the height of each layer and $Q_i$ is temperature or relative humidity from the radiosonde data at the corresponding height. The MB and RMSE were used to evaluate the deviation between the retrieval parameter and sounding for each height layer. The correlation coefficient indicates the difference between each retrieval profile, organized by the 47 height layers and corresponding sounding data.





The 382 groups of sounding data from 2014 to January 2015, excluding rainy conditions, were used as independent datasets in the analysis of the accuracy of the retrieval profiles by the three methods, the results were shown in Figure 7.

Figure7 shows that, in the retrieval of temperature, the MB of BPNN(Cloud) fluctuated between -0.9K and 0.6 K, while the RMSE did so from 0.2 K to 3.2 K. In addition, the MB of BPNN(No-cloud) fluctuated between −1K and 1.1K and the RMSE increases from 0.5 K to 3.6 K over the height range of 0-10 km. The MB of the MWR changed substantially between -1K and -6K, while the RMSE increases from 0.3 K to 6.5 K, with a more rapid increase above 7 km.

In the retrieved humidity profiles, the MB of BPNN(Cloud) fluctuated between -6% and 7%, and the RMSE increased rapidly from 2.5% to 24% from the ground to 6 km, and was 6%-22% above 6km. The MB of BPNN(No-cloud) fluctuated between -9% and 7%, while the RMSE had already increased to 27% at 3.5 km and oscillated between 15% and 27% from 2.5 km to 10 km. The MB of the MWR results was 1%-15% while the RMSE increased to 31% from 0 to 3.5 km, and then fluctuated down to 22.5% in oscillations at 10 km.

The RMSE statistics showed that, the temperature profile roughly conforms the characteristics of the ground-based remote sensing. The retrieval accuracy gradually reduced from the ground to 10km. But the humidity profiles have large difference with the increasing of the height. In order to analyse the impact of cloud more clearly, we count the RMSE between BPNN(Cloud) and BPNN(No-cloud) separately according to the difference of the height of cloud-base. The cloud samples were devided into three categories: low-cloud, mid-cloud and high-cloud, following the work of (China Meteorological Administration, 2007). The low-cloud was defined when the cloud-base under 2.5km, and the mid-cloud was 2.5-4.5km, meanwhile the high-cloud was above 4.5km. The number of low-cloud samples was 70, the mid-cloud was 63, and the high-cloud was 62. The results were shown in Figure 8.

As the comparison for humidity profiles of different cloudy condtions shown in Figure 8, the RMSE increased obviously at the height between 1.5km and 5km in low-cloudy condition, but in mid-cloudy condition, the maximum RMSE values appeared at about 5km, meanwhile RMSE at above 5km in high-cloudy condition was obviously larger than the other two conditons. After the addition cloud information, BPNN(Cloud) had clearly improvement at above cloud layers, which it can not change the RMSE trend with the height increasing. So it indicated that the presence of cloud could cause obvious error at cloud and above cloud height, the addition cloud information can amend the error, but can not eliminate it.

In order to quantify the three retrieved profiles, we also selected randomly 75 samples of the cloudy condition to comparing by correlation coefficients with the corresponding sounding data, The results were shown in Figure 9.

For retrieved temperature profiles, the average correlation coefficient between BPNN(No-cloud) and the radiosonde data was 0.990 and that between BPNN(Cloud) and the radiosonde data was 0.994. In the whole 75 group samples, the number of correlation coefficients generated increased to 49 after the addition of cloud information. The average correlation coefficient between the MWR and radiosonde was 0.992.

For the relative humidity profiles, the average correlation coefficient between BPNN(No-cloud) and the radiosonde data was 0.685, and while that between BPNN(Cloud) and the radiosonde data was 0.805. After the addition cloud information, 49 out of the 75 samples showed improved correlation. The maximum increase in the correlation coefficient was 0.330, and the average correlation coefficient between the MWR and radiosonde was 0.657.



## 5 Advantage of the MWCR

The results from the four typical cases demonstrate that the temperature and humidity profiles retrieved by the MWR show almost the same trends as the radiosonde data, but do not show a good response in layers with cloud present. Furthermore, the MB and RMSE statistics indicate that the deviation increases significantly above 3 km, but the retrieval by the BPNN with added cloud parameters is much better, especially as it can reflect the feature that relative humidity increases significantly in cloudy layers.

To analyze the reasons for this, we next compare data from MWCR and the cloud-base height observed by the far infrared radiometer components that configured the MWR during the entire process of the cloud corresponding to the four aforementioned cases(Figure 10). It should be noted that the cloud base height of the far-infrared component is set to 10 km when it does not detect cloud.

Figure 10 shows that, in case b, the cloud base height obtained by far-infrared remote sensing is close to that from the cloud radar. However, in cases a, c and d, there is a considerable difference between the cloud parameters measured by the two instruments. There was slight fog or haze at the sounding time of cases a and c, and the cloud-base height obtained by far-infrared was measured at 1 km or less at the time. Meanwhile, there were thin cloud conditions at the sounding time of case d, but the far-infrared did not detect the existence of cloud.

When there are high levels of fog, haze or relative humidity at the surface, the MWR cannot detect the existence of high cloud because of the weather, and the obtained cloud base height may show large deviation or considerable error. For thin cloud, there may also be missing data in the MWR dataset. However, according to section 4-a, the existence of cloud layers can change the BT measured by the MWR, and thus the cloud-base height is insufficient for the process of retrieving atmospheric profiles. Moreover, the measurement of cloud-base height can sometimes show large deviation, indicating that large errors exist in the retrieval of profiles in cloudy conditions.

In contrast, when using a combination of MWR and cloud radar observations, the MWCR uses cloud particle scattering properties of the electromagnetic waves. We can analyze the radar echo to obtain the various features of the cloud, which can reflect the cloud's macroscopic and microscopic structure. From this, we can obtain the cloud-base height more accurately. Cloud radar can also obtain the cloud thickness, cloud cover and even microphysical parameters in the cloud. This can provide a more complete set of cloud parameter information for the retrieval process. However, because of the short time period of combined observations, the amount of cloud radar data is limited, so the cloud-base height and cloud thickness in a part of the training sample need to be judged via the sounding data. Unfortunately, in this study, the radiosonde was unable to provide accurate cloud information, with the exception of cloud base height and cloud thickness; therefore, we were only able to add two channels (cloud height and thickness) in the final retrieval. We aim to solve this problem in future research.

## 6 Conclusions

This study used a combination of active and passive remote sensing techniques to tackle the problem of increased bias in



MWR retrievals during cloudy conditions. The cloud-base height and cloud thickness measured by the MWCR were applied in the process of retrieving atmospheric profiles. We compared the retrieved profiles with and without cloud information and the product of a radiometer with sounding data. To analyze the resulting bias, we compared the cloud radar data with the cloud-base height using the far-infrared radiometer of the MWR at the same time. The key conclusions can be summarized as follows:

1. The three retrieval methods basically showed consistent results in terms of their error trends for temperature and humidity profiles with height change. The accuracy of retrieval was high near the ground and decreased with height. This confirms the detection performance of the ground-based MWR.

2. The comparison between the MWCR and the far-infrared radiometer components configured with the MWR itself showed that the latter obtained the cloud-base height, but struggled to measure this parameter steadily in the long-term because the data quality was negatively affected by the weather conditions.

3. The precision of the retrieval without cloud and the MWR product reduced significantly in the cloudy layers, but the precision significantly improved after the addition of accurate information from the cloud radar. Compared with retrieval without cloud information, the retrieved temperature and humidity profiles were closer to the radiosonde data after cloud information was added for the cloudy layers. This improvement was especially clear in the upper layers.

This study analyzed the causes of increased deviation detected by the MWR during periods of cloud, and then combined MWCR measurements to improve the data. This verified that the use of accurate cloud information to improve the retrieval accuracy is a feasible approach. In future experiments, we will combine cloud radar and MWR long-term observations and use more abundant cloud distribution information to improve the retrieval through the BT of the MWR and obtain a more complete vertical profile.

**Acknowledgements.** The research has reveived funding from the key program of 2015Z001 of CAMS. The authors would like to acknowledge and thank Li Liang, Xiaohu Pu, and Fa Tao for their work in improving the manuscript.



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





**Table 1.** Statistical comparison of simulated BT and MWR measurements.

| Frequency/GHz | Mean Deviation/K | Standard Deviation/K | Correlation coefficient |
|---|---|---|---|
| 22.24 | 2.145 | 2.508 | 0.984 |
| 23.04 | 1.988 | 2.338 | 0.984 |
| 23.84 | 1.628 | 2.003 | 0.983 |
| 25.44 | 1.213 | 1.662 | 0.976 |
| 26.24 | 1.065 | 1.498 | 0.974 |
| 27.84 | 0.991 | 1.477 | 0.963 |
| 31.40 | 1.084 | 1.672 | 0.937 |
| 51.26 | 1.590 | 2.403 | 0.900 |
| 52.28 | 1.298 | 1.913 | 0.909 |
| 53.86 | 0.495 | 0.680 | 0.967 |
| 54.94 | 0.350 | 0.397 | 0.988 |
| 56.66 | 0.360 | 0.463 | 0.987 |
| 57.30 | 0.396 | 0.513 | 0.985 |
| 58.00 | 0.455 | 0.574 | 0.981 |





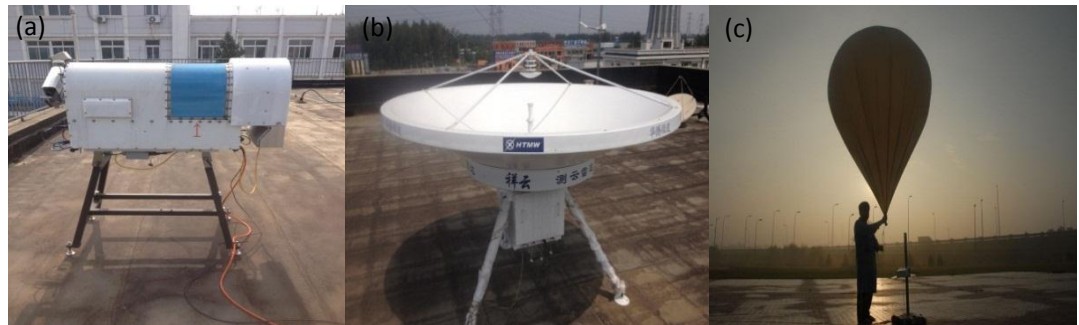

**Figure 1.** Photographs of the (a) RPG-HATPRO MWR;(b) cloud radar;and (c) sounding balloon.





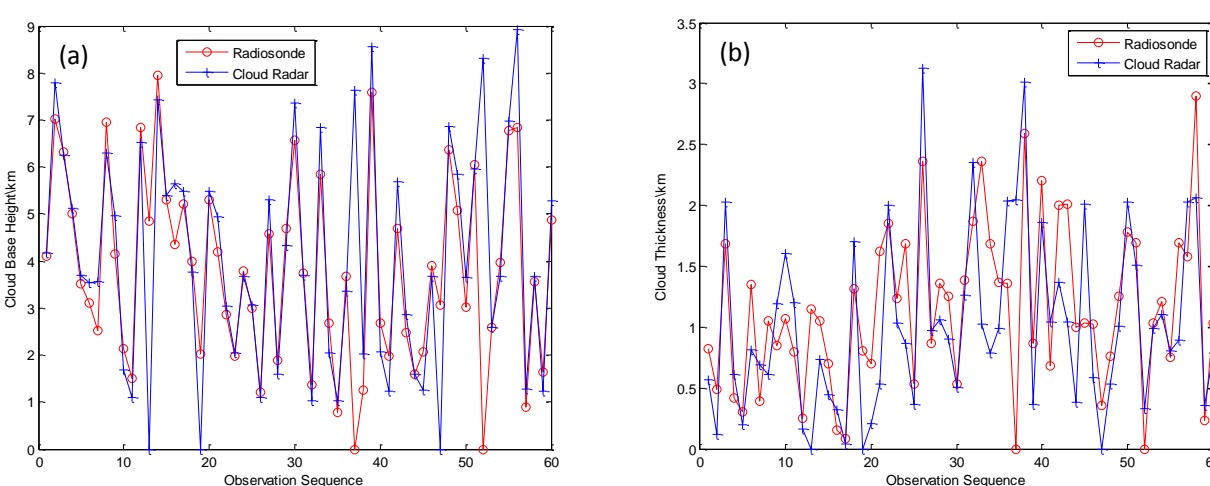

**Figure 2.** Comparison of cloud information: (a) cloud base height (km), (b) cloud thickness (km).





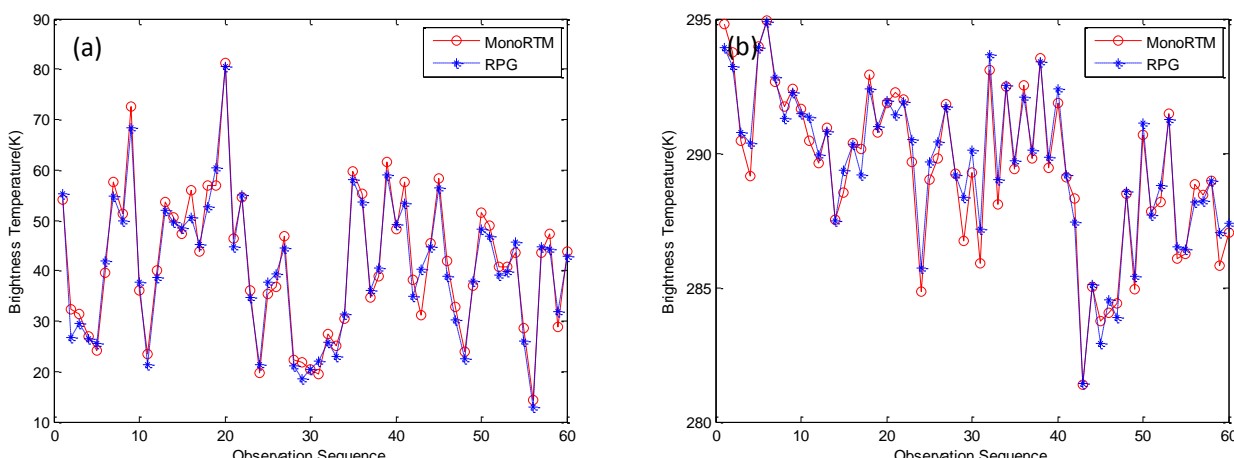

**Figure 3.** Comparison of MonoRTM-derived BT and observation by MWR in the (a) 22.24 GHz channel; and (b) 58 GHz channel.



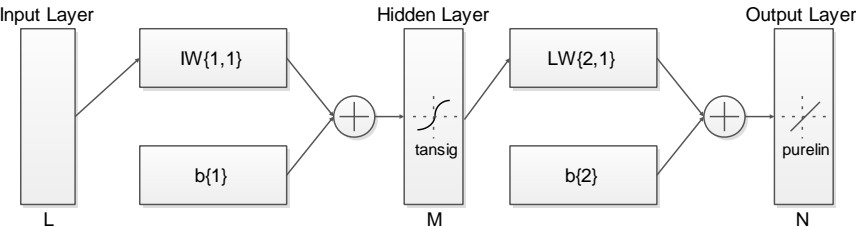

**Figure 4.** Schematic diagram of BPNN.




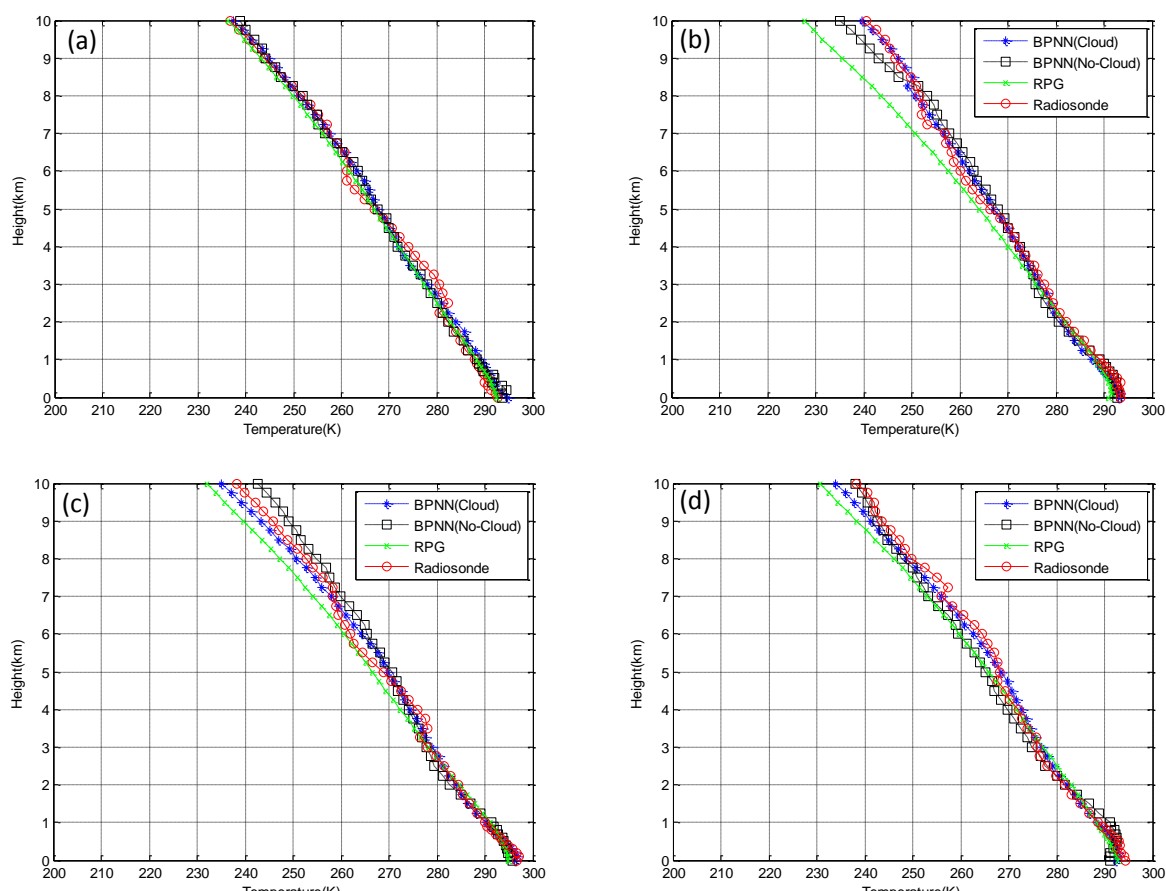

**Figure 5.** Comparison between BPNN(Cloud), BPNN(No-cloud), the MWR and the radiosonde for temperature profiles at (a) 23:15 UTC 06 Nov, 2014;(b) 23:15 UTC 03 Oct, 2014; (c) 23:15 UTC 02 Nov, 2014; and (d) 11:15 UTC 08 Oct, 2014.





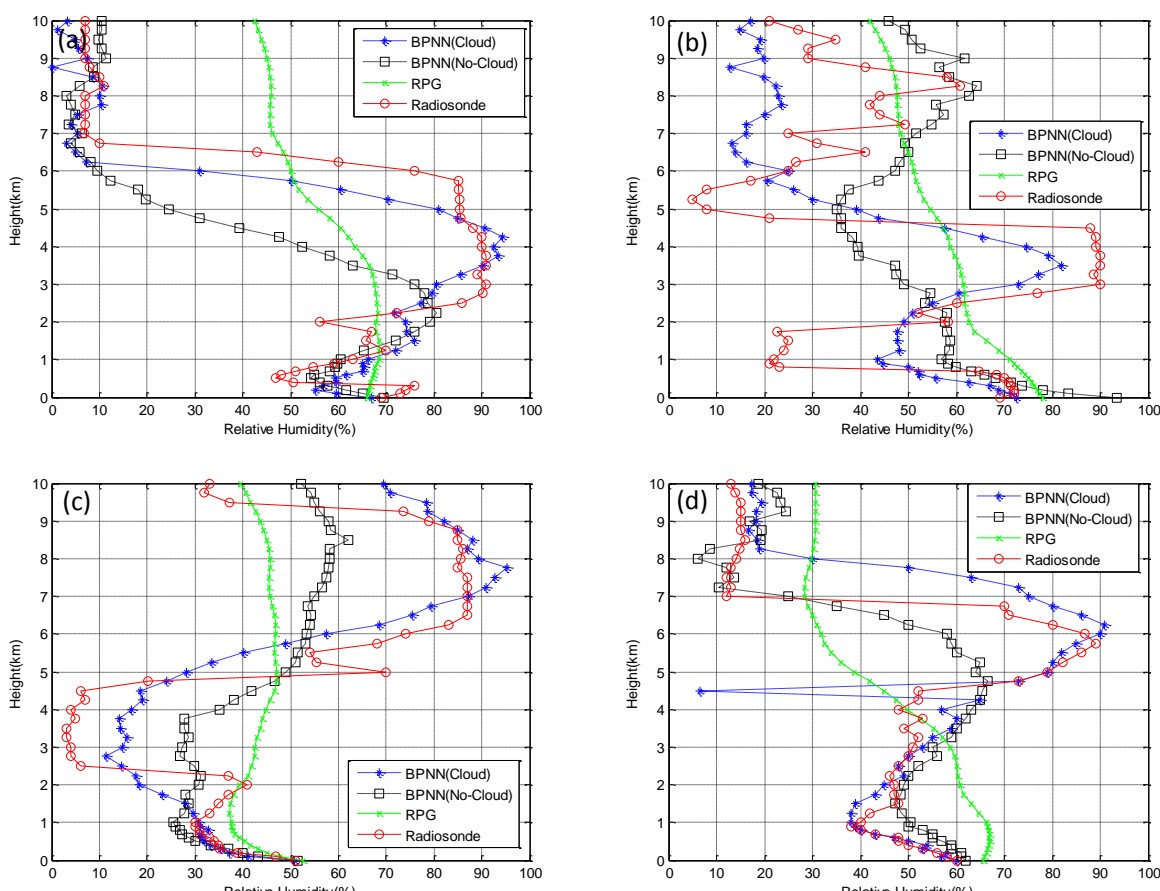

**Figure 6.** Comparison between BPNN(Cloud), BPNN(No-cloud), the MWR and the radiosonde for relative humidity profiles at (a) 23:15 UTC 06 Nov, 2014; (b) 23:15 UTC 03 Oct, 2014, (c) 23:15 UTC 02 Nov, 2014; and (d) 11:15 UTC 08 Oct, 2014.




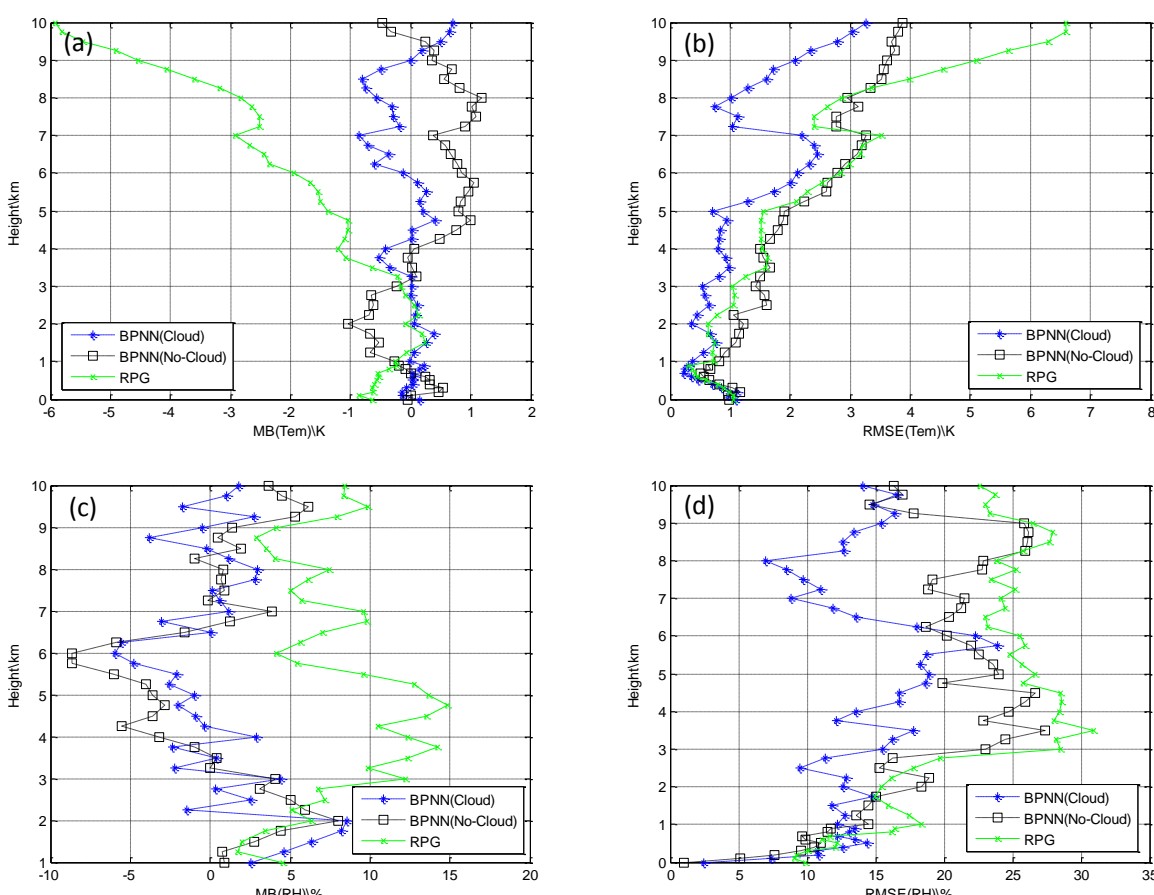

**Figure 7.** Statistical results between BPNN(Cloud), BPNN(No-cloud), the MWR and the radiosonde :including (a)MB for temperature profiles; (b) RMSE for temperature profiles; (c) MB for humidity profiles; and (d) RMSE for humidity profiles.





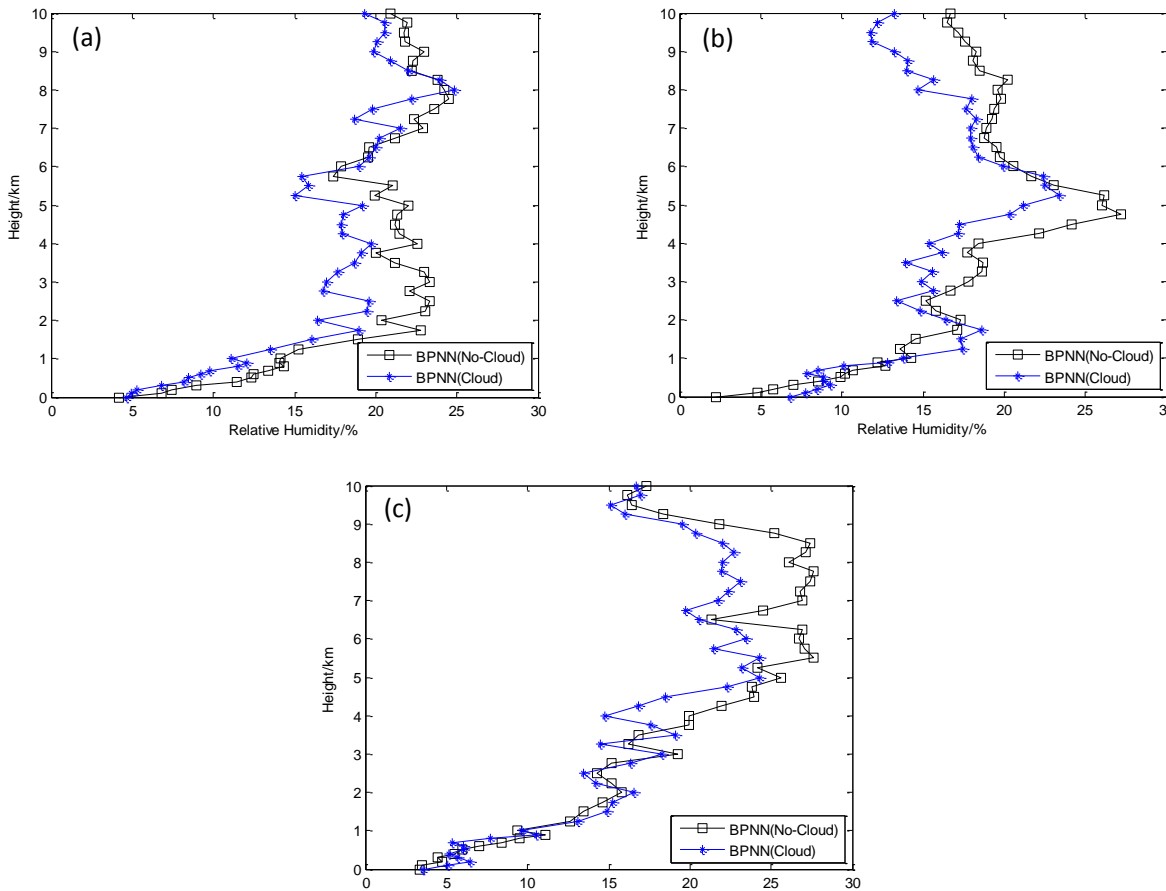

**Figure 8.** RMSE comparison for humidity profiles between BPNN(Cloud), BPNN(No-cloud) in different cloudy conditions: (a) low-cloud; (b) mid-cloud; and (c) high-cloud.





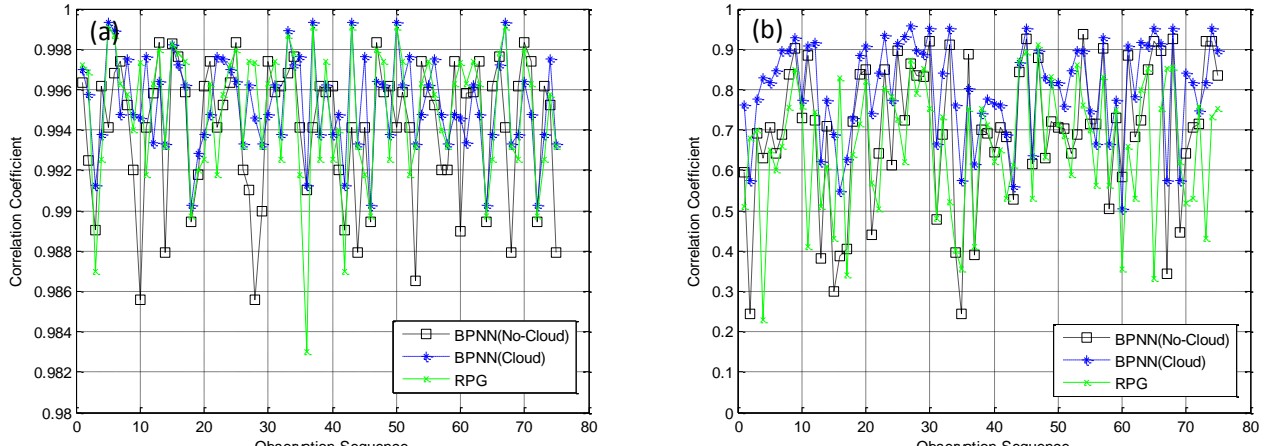

**Figure 9.** Comparison of the correlation coefficient between BPNN(Cloud), BPNN(No-cloud), the MWR and the radiosonde for (a) temperature profiles; and (b) relative humidity profiles.





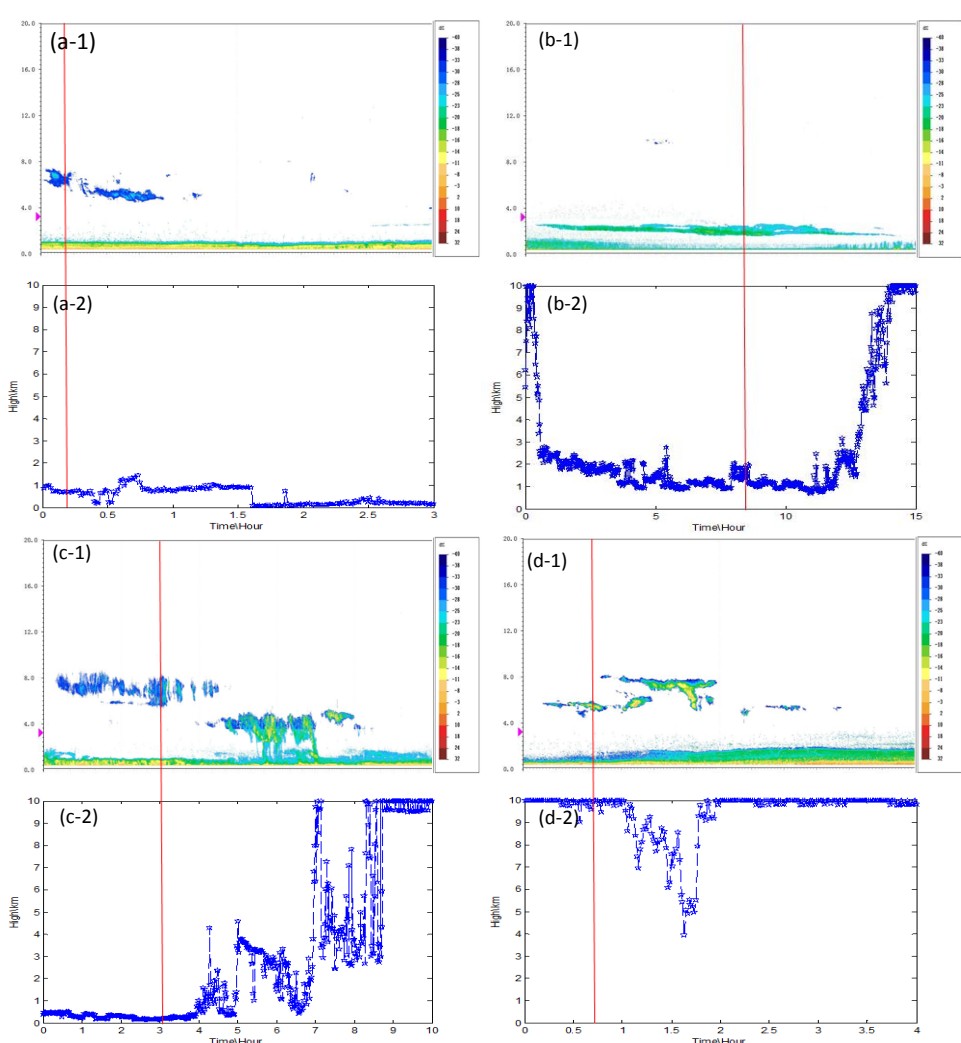

**Figure 10.** Comparison for cloud radar the (a1-d1) and the (a2-d2) far infrared radiometer components: (a) 06 Nov, 2014; (b) 03 Oct 2014; (c) 02 Nov, 2014; (d) 03 Oct, 2014. The red line represents the release time of the sounding ballon.