# Peer review of "Research on Retrieval of Atmospheric Temperature and Humidity Profiles from combined Ground-based Microwave Radiometer and Cloud Radar Observations"

_Atmospheric Measurement Techniques, 2016_

## Referee Comment (RC1) · Anonymous Referee #1 · 11 Oct 2016

The paper presents a method to retrieve temperature and relative humidity profiles by using microwave radiometer, cloud radar and radiosonde data. In order to retrieve the atmospheric parameter, a neural network approach is used. The topic is of high interest for the scientific community and fits well the scope of the journal.

Nonetheless, I cannot recommend the paper in its current form for publication on AMT. I identify some general problems, which are discussed in the following.

First, the background knowledge and state of the art description is very poor, which leads to an imprecise contextualization of the work. If the authors claim to present an

improved method for the retrieval of thermodynamic profiles, they should also provide a more extensive overview of the different methodologies applied so far.

In addition, there is in general a strong lack of references. Continuously, the authors state affirmations but no source is cited. For example: page 2, line 5; page 3, line 5; page 7, line 25-26; page 7, line 18, etc.

Moreover, the scientific methodology is often neither clear nor precise. Strong assumptions and/or simplifications are performed, e.g. the calculation of the liquid water content from relative humidity, the cloudy/clear detection and cloud geometry estimation from relative humidity, etc. Those simplifications are not completely justified and/or discussed.

Also, there is important information missing in the description of the algorithm and instrumentation used in the retrieval. An example of this is section 3.2, which aims to provide an explanation of the neural network method applied. Here, a description of "what is indeed a neural network algorithm" or references to another source explaining it are missing. Because of that, many points remain incomplete, e.g.: what is a layer, why using 3 layers, what is a hidden layer or why using a tansig transfer function. Another example is the reduced description of the instrument used in the study. I would encourage the authors to work also on this part and cite useful references like Rose et al. 2004, which provides a complete and detailed description of the HATPRO instrument.

Finally, I personally would expect the use of the language to be more accurate: non-scientific opinions are used frequently. For example, in line 16 in page 6: I would not say they are the most popular methods. Other methods, e.g. iterative methods such as optimal estimation, are widely used in the scientific community. Indeed papers using the later are cited by the authors in the introduction.

For the reasons discussed above, I think that the paper is not mature enough to be published in its current form and thus I recommend its rejection.

References: Rose, T., S. Crewell, U. Löhnert, and C. Simmer, 2005: A network suitable microwave radiometer for operational monitoring of the cloudy atmosphere. Atmos. Res., 75, 183–200, doi:10.1016/ j.atmosres.2004.12.005.

———————————————————

---

## Referee Comment (RC2) · Anonymous Referee #2 · 19 Oct 2016

General comments:

This study quantifies how utilizing cloud base and cloud thickness estimates from MWCR and radiosonde data is beneficial for improving temperature and humidity retrievals derived from MWR measurements. In general, the methodology is unclear. There is decent discussion of the retrievals and the various measurements but it is unclear how the BT and cloud information derived from the radiosonde data is used in the retrieval process. A reader should be able to read a clear, consolidated description of the methodology stating how the retrievals were developed.

[Figure]

One of the key findings is that adding cloud macrophysical measurements improves the retrieved temperature and humidity profile accuracy. While it is useful to quantify this truth, the paper needs to put the current study in context with previous findings. How this approach is different or better than previous retrievals (more references are needed to put this paper in context) will help make clear the purpose of the paper.

In addition, adding the location of the study will clarify in what environment these measurements were taken and suggest how applicable they are to other locations.

These changes require major revisions.

Specific Comments:

Section 2 should include more detail regarding the measurements. Was the MWR calibrated via a tip-curve technique, LN2 calibrations, or a combination of both? What is the threshold of the MWCR reflectivity (in dBz) to detect cloud base height and cloud thickness? How does changing this threshold affect the comparison in Fig 2?

Section 2.2 discusses the "pre-treatment of sounding data". Why is the simulated BT not included in this section? It would make sense to discuss all the measurements and radiosonde-derived parameters together before introducing the retrieval methodology.

In table 1 what is the average bias of the 60 samples of each MWR channel compared to the radiosonde-derived BT value? Bias of the MWR BT will affect the accuracy of the temperature retrievals.

Section 3 is titled "Retrieval methodology" and is where I expect to find clear explanation of the retrieval processes. Yet, prior to this section (pg 3, line 14-19) a training dataset is introduced, which is based on the sounding data. It seems that the description of the training data set should be included in the methodology section and the description of the data sources should be introduced prior to this section. I do not know what is meant by: "Initially, the pre-treatment of the radiosonde data focused on the removal of rainy or uncertain weather conditions" (pg 3, 15-16). There needs to be

a clear, consolidated explanation of the retrieval and the training dataset used in the retrieval.

Page 5 line 16: "The linear regression method and neural networks are probably the most popular in current research." The most popular what? Provide references. What about optimal estimation techniques?

Section 3.2.2 is where the input elements are defined for "no-cloud" and "cloud", yet the retrieval nomenclature is introduced on page 8 lines 14-15. Section 3.2.2, called "Methodology on the comparison of the addition of cloud", actually introduces both the "no-cloud" and "cloud" retrievals but fails to label them leaving the reader to decipher that these two methods are what will be later and be dubbed BBPN(cloud) and BPNN(no-cloud).

Generally, the methodology should explicitly describe the training data set, the two BPNN retrievals, and the HATPRO retrieval so it is clear what is being compared in figures 5-9.

Why aren't the same soundings to train the RPG-HATPRO retrieval (page7 lines 6-7) and were used to train the BBPN retrievals? The differences in the results section could be due to differences in the training set and not the retrieval methods.

What is meant by "In the whole 75 group samples, the number of correlation coefficients generated increased to 49 after the addition of cloud information."? (page9, lines 29-30)

Section 4.1 "Theory" This section seems out of place as well. The information on page 7 could be condensed and used to motivate why you did the retrievals. By the time the reader gets to page 8, lines 2-4, you have already described the retrievals, thus you are motivating why you are doing the new retrieval after you have already described it in a fragmented way.

Page 7 lines 24-25. Cite the research that has shown this statement.

Page 10 line 28: reword the last paragraph of section 5 (page 10 lines 22-31). The use of "can" makes it difficult to discern what was actually done and what will be done in future. On line 28 "judged" should be reworded because the sounding data is not used to judge the cloud-base height and thickness. Perhaps use "performed"?

Stay consistent in referring to the MWR retrieval. The Figures are labeled "RPG" and the text talks of MWR retrievals, but it is unclear if the MWR retrievals are RPG derived or the BPNN technique. For example: page 11, line 12 says "3. The precision of the retrieval without cloud and the MWR product reduced significantly in the cloud layers, but the precision significantly improved after addition of accurate information from the cloud radar." This sentence seems to be the main message of the paper yet it is still unclear which retrievals are being referred to.

Overall the language needs to be more precise. Avoid statements such as those used in the conclusion: "basically showed", ". . . tackle the problem of increased bias . . .", or " . . . a more complete vertical profile. "

Technical corrections:

Page1, line 4: Should be "Key Laboratory for Cloud Physics,"

Reword: pg 4 line 16-17." . . . the cloud information must be more accurate and richer than before." This is unclear.

"Clean-sky" should be "clear-sky" throughout the manuscript.

Fig 10: The axis labels are too small. Perhaps make it a 4 panel figure (a-d) with infrared radiometer components overlaid the cloud radar reflectivities.
* * *

---

## Author Comment (AC1) · 21 Dec 2016

Anonymous Referee#1:

Response: I am very grateful that the referee gave me a lot of valuable advice, the following is my understand and reply, the black font is the comments from Referees, and the red font is my reply, please review it.

Comment: The paper presents a method to retrieve temperature and relative humidity profiles by using microwave radiometer, cloud radar and radiosonde data. In order to retrieve the atmospheric parameter, a neural network approach is used. The topic is of high interest for the scientific community and fits well the scope of the journal.
Nonetheless, I cannot recommend the paper in its current form for publication on AMT. I identify some general problems, which are discussed in the following.
First, the background knowledge and state of the art description is very poor, which leads to an imprecise contextualization of the work. If the authors claim to present an improved method for the retrieval of thermodynamic profiles, they should also provide a more extensive overview of the different methodologies applied so far.

Response: I have to agree that the background knowledge is poor, According to the literature I found, In the past researches, retrieval of cloud liquid water content from combined MWR and radar is more attention, and for the improvement of humidity profiles,  more research combine MWR and wind profiler radar. I have added the content of "introduction", please review it.

Comment: In addition, there is in general a strong lack of references. Continuously, the authors state affirmations but no source is cited. For example: page 2, line 5; page 3, line 5; page 7, line 25-26; page 7, line 18, etc.

Response: This is my fault, I have add the reference on the revise manuscript.

Comment: Moreover, the scientific methodology is often neither clear nor precise. Strong assumptions and/or simplifications are performed, e.g. the calculation of the liquid water content from relative humidity, the cloudy/clear detection and cloud geometry estimation from relative humidity, etc. Those simplifications are not completely justified and/or discussed.

Response: It is no doubt that this manuscript used some assumptions, but we have to assumpt it by referring to other studies. Following is my reply:
There are some uncertainty in the manuscript:
1. Cloud base height and cloud thickness estimated by relative humidity from radiosonde;
2. liquid water content calculation from relative humidity;
3. The deviation between BT calculated by MonoRTM and BT measured by radiometer.
But in this experiment, the training dataset must have the cloud parameters corresponding with every radiosonde profile. The cloud base height and cloud thickness have to be estimated by radiosonde data, because the samples of cloud

detection by cloud radar are very limited, we could not get nearly a decade of detecting cloud data corresponding the radiosonde. And this part of the study was referred by previous studies{wang and Rossow 1995 Journal of applied meteorology} (Wang and Rossow,1995) have proposed a method which us rawinsonde data to estimate cloud vertical structure, maximum relative humidity in a cloud of at least 87%,minimum relative humidity of at least 84%, and relative humidity jumps exceeding 3% at cloud-layer top and base. Cai et.al 2014 have described that setting the relative humidity at 81% as the threshold, the TS score is 0.66 as the highest score. Thought observation, relative humidity from radiosonde increases rapidly on the cloud layer, the error discriminating cloud base and thickness is small caused by the relative humidity of 84% as the threshold. In order to verify this method, I compared the cloud from radiosonde with the result of cloud radar. The 21 samples comparison is actually little, so I increased the number of samples comparison in the revised manuscript. It is undeniable that this approach have some error, I should be improve it in the follow-up work.

The profiles of cloud liquid water as required in the calculation of brightness temperatures are not available in the conventional upper-air ascent data, so some researchers{Tan et al 2010} assume it by relative humidity. This method have described in Wang et al(1995){Wang Z. P. 1995 Simulation of atmospheric vapor, liquid water content, and excess propagation path length based on 3-channel microwave radiometer sensings    J.Nanjing Inst. Meteor}. In the current condition, I can not verify this part very clearly as the liquid water content could not be measured as the standard. And the liquid water content estimated is used to the MonoRTM, so I compared the result of MonoRTM with the measured by microwave radiometer.

With the increasing of the radiometer and cloud radar samples, this part will be greatly improved.

Hope my reply could let experts satisfied.

Comment: Also, there is important information missing in the description of the algorithm and instrumentation used in the retrieval. An example of this is section 3.2, which aims to provide an explanation of the neural network method applied. Here, a description of "what is indeed a neural network algorithm" or references to another source explaining it are missing. Because of that, many points remain incomplete, e.g.: what is a layer, why using 3 layers, what is a hidden layer or why using a tansig transfer function. Another example is the reduced description of the instrument used in the study. I would encourage the authors to work also on this part and cite useful references like Rose et al. 2004, which provides a complete and detailed description of the HATPRO instrument.

Response: I'm sorry that the description of the algorithm and the instrument may be not clear. The neural network algorithm has been clearly described in the reference,
In the revised manuscript, I have added the references and the description of the network.
I have read Rose et al. 2004 and some other reference about the HATPRO, in the

revised manuscript, I increased the description and references of the instrument.

Comment: Finally, I personally would expect the use of the language to be more accurate: nonscientific opinions are used frequently. For example, in line 16 in page 6: I would not say they are the most popular methods. Other methods, e.g. iterative methods such as optimal estimation, are widely used in the scientific community. Indeed papers using the later are cited by the authors in the introduction.

Response: This is my neglect, I have revised this part, and modified the language described in the full manscript, please review it.

Comment: For the reasons discussed above, I think that the paper is not mature enough to be published in its current form and thus I recommend its rejection.

---

## Author Comment (AC2) · 21 Dec 2016

Anonymous Referee#2:

I am very grateful that the referee gave me a lot of valuable advice, the following is the comment and response.

Comment: This study quantifies how utilizing cloud base and cloud thickness estimates from MWCR and radiosonde data is beneficial for improving temperature and humidity retrievals derived from MWR measurements. In general, the methodology is unclear.

There is decent discussion of the retrievals and the various measurements but it is unclear how the BT and cloud information derived from the radiosonde data is used in the retrieval process. A reader should be able to read a clear, consolidated description of the methodology stating how the retrievals were developed.

Response: The retrieval method based on the neural network algorithm. Ensure that the other conditions unchanged, and the cloud base height and cloud thickness were set into the input layers as the independent elements. And the retrieval results are compared with the retrieval results without the cloud base height and cloud thickness data.

The description may be not clear, I have revised it, please review.

Comment: One of the key findings is that adding cloud macrophysical measurements improves the retrieved temperature and humidity profile accuracy. While it is useful to quantify this truth, the paper needs to put the current study in context with previous findings. How this approach is different or better than previous retrievals (more references are needed to put this paper in context) will help make clear the purpose of the paper.

Response: I realized the reported literature is really poor, I have modified this part. In the revised manuscript, the source and development of the neural network algorithm were described. The overview of the different instrument joint detect atmospheric profiles have been shown, and the part of link of this study with previous studies is added, please review it.

Comment: In addition, adding the location of the study will clarify in what environment these measurements were taken and suggest how applicable they are to other locations.

Response: The location of the study is nanjiao, Beijing. In the current study, I don't think this method could be suitable at the place which is high humidity on the ground, because the estimated cloud by radiosonde may be large error. I think this method should be applicable at the other environment.

Comment: These changes require major revisions.

Specific Comments:

Section 2 should include more detail regarding the measurements. Was the MWR calibrated via a tip-curve technique, LN2 calibrations, or a combination of both? What

is the threshold of the MWCR reflectivity (in dBz) to detect cloud base height and cloud thickness? How does changing this threshold affect the comparison in Fig 2?

Response: In the revised manuscript, I add the detail about the RPG-HATPRO. The MWR calibrated via a combination of tip-curve technique and LN2 calibrations. The threshold of the MWCR reflectivity is -30dBz, the setting of threshold is fixed value,I'm sorry that I have not change it.

Comment: Section 2.2 discusses the "pre-treatment of sounding data". Why is the simulated BT not included in this section? It would make sense to discuss all the measurements and radiosonde-derived parameters together before introducing the retrieval methodology.
In table 1 what is the average bias of the 60 samples of each MWR channel compared to the radiosonde-derived BT value? Bias of the MWR BT will affect the accuracy of the temperature retrievals.

Response: I have revised. The simulated BT is included in "pre-treatment of sounding data". In the table 1, I added the average bias of the MWR and radiosonde-derived BT.

Comment: Section 3 is titled "Retrieval methodology" and is where I expect to find clear explanation of the retrieval processes. Yet, prior to this section (pg 3, line 14-19) a training dataset is introduced, which is based on the sounding data. It seems that the description of the training data set should be included in the methodology section and the description of the data sources should be introduced prior to this section. I do not know what is meant by: "Initially, the pre-treatment of the radiosonde data focused on the removal of rainy or uncertain weather conditions" (pg 3, 15-16). There needs to be a clear, consolidated explanation of the retrieval and the training dataset used in the retrieval.

Response: I have revised that "simulated BT based on MonoRTM" is included in "pre-treatment of sounding data" now. And "Retrieval methodology" should be only the description of neural network. I added the explanation of the way to adding cloud information. please review.

Comment: Page 5 line 16: "The linear regression method and neural networks are probably the most popular in current research." The most popular what? Provide references. What about optimal estimation techniques?

Response: This is my neglect, I'm sorry for this expression and I will delete it, please review it.

Comment: Section 3.2.2 is where the input elements are defined for "no-cloud" and

"cloud", yet the retrieval nomenclature is introduced on page 8 lines 14-15. Section 3.2.2, called "Methodology on the comparison of the addition of cloud", actually introduces both the "no-cloud" and "cloud" retrievals but fails to label them leaving the reader to decipher that these two methods are what will be later and be dubbed BBPN(cloud) and BPNN(no-cloud).

Response: This is my neglect, BPNN(cloud) and BPNN(no-cloud) should be defined in the "Methodology on the comparison of the addition of cloud", I have revised, please review.

Comment: Generally, the methodology should explicitly describe the training data set, the two BPNN retrievals, and the HATPRO retrieval so it is clear what is being compared in figures 5-9.
Why aren't the same soundings to train the RPG-HATPRO retrieval (page7 lines 6-7) and were used to train the BBPN retrievals? The differences in the results section could be due to differences in the training set and not the retrieval methods.

Response:
The description of the two BPNN retrievals, and the RPG retrieval have been added .
It is no doubt that the same soundings to train the RPG retrieval could be more meaning. But we can not modify the RPG retrieval, The manufacturer have finished the process of retrieval when we get the instrument, and we only get the matrix of retrieval. But I still think it is necessary to add RPG retrieval to compare, because if the RPG retrieval have good performance, the work which I study is meaningless.

Comment: What is meant by "In the whole 75 group samples, the number of correlation coefficients generated increased to 49 after the addition of cloud information."? (page9, lines 29-30)

Response: I want to show that 49 group samples have been better when adding cloud information, while the number of whole samples is 75. I have deleted and revised this expression in the revised manuscript.

Comment: Section 4.1 "Theory" This section seems out of place as well. The information on page 7 could be condensed and used to motivate why you did the retrievals. By the time the reader gets to page 8, lines 2-4, you have already described the retrievals, thus you are motivating why you are doing the new retrieval after you have already described it in a fragmented way.
Response: In this section, I want to explain the theory of the adding cloud information, so the description of the retrieval will be deleted in this section, and the theory need to be condensed, please review.

Page 7 lines 24-25. Cite the research that has shown this statement.
Page 10 line 28: reword the last paragraph of section 5 (page 10 lines 22-31). The use

of "can" makes it difficult to discern what was actually done and what will be done in future. On line 28 "judged" should be reworded because the sounding data is not used to judge the cloud-base height and thickness. Perhaps use "performed"?

Response: I have reworded the use of "can", and The "judged" is substituted by "estimated", please review.

Comment: Stay consistent in referring to the MWR retrieval. The Figures are labeled "RPG" and the text talks of MWR retrievals, but it is unclear if the MWR retrievals are RPG derived or the BPNN technique. For example: page 11, line 12 says "3. The precision of the retrieval without cloud and the MWR product reduced significantly in the cloud layers, but the precision significantly improved after addition of accurate information from the cloud radar." This sentence seems to be the main message of the paper yet it is still unclear which retrievals are being referred to.

Response: The description may be not clear, "MWR product" is same as "RPG retrieval", but In the manuscript, I should stay consistent in referring to retrieval, I will revise it at once.

Overall the language needs to be more precise. Avoid statements such as those used in the conclusion: "basically showed", ". . . tackle the problem of increased bias . . .", or " . . . a more complete vertical profile.

Response: I'm sorry for the language, I will check the whole manuscript and avoid statements like this.

Comment: Technical corrections:
Page1, line 4: Should be "Key Laboratory for Cloud Physics,"
Response: I have revised it, please review.
Comment: Reword: pg 4 line 16-17." . . . the cloud information must be more accurate and richer than before." This is unclear. "Clean-sky" should be "clear-sky" throughout the manuscript.
Response: I'm very sorry for this error, I have revised it, please review.
Comment: Fig 10: The axis labels are too small. Perhaps make it a 4 panel figure (a-d) with infrared radiometer components overlaid the cloud radar reflectivities.